# Layout Guidelines for 3D Printing Devices

**Arkadiusz Kowalski** [1] **and Robert Waszkowski** [2,*]

1   Faculty of Mechanical Engineering, Wrocław University of Science and Technology, 50-371 Wrocław, Poland; arkadiusz.kowalski@pwr.edu.pl
2   Cybernetics Faculty, Military University of Technology, 00-908 Warszawa, Poland
*   Correspondence: robert.waszkowski@wat.edu.pl

**Abstract:** 3D printing methods are constantly gaining in popularity among investors, allowing for the production of products with a complex structure, and also used in the production of products with increasingly longer production series. It is planned to build factories (or those already under construction) in which 3D printing devices are the basic production devices. It is therefore important to develop guidelines and recommendations for layout design principles for additive technologies' devices. A question should be asked: will the development of a layout for additive technology machines in future factories differ from the preparation of a layout plan, for example, removal machining devices? Is it safe to assume that the mathematical methods of optimizing the layout of Computerized Relative Allocation of Facilities Technique (CRAFT), Computerized Relationship Layout Planning (CORELAP), and Modified Spanning Tree (MST) workstations or Schmigalla triangles will also work for 3D printing machines and devices? In search of answers to these questions, the article will attempt to apply a selected mathematical method to optimize the layout of workstations when machines and devices of additive technology are deployed for the assumed technological process and implemented according to frequently used Selective Laser Sintering (SLS) and Selective Laser Melting (SLM) technologies. A sample layout will be prepared for the assumed production plan and selected 3D printing technologies. Requirements and guidelines relevant to the development of layout plans will be collected regarding the necessary space, installations, and connections.

**Keywords:** layout; 3D printing devices; methods of optimizing the arrangement of workstations

## 1. Introduction

With the development of 3D printing methods, factories equipped with this type of equipment are increasingly being built. It is important from the investor's point of view that the developed layout plan takes into account the specific requirements of 3D printing machines. When developing the above-mentioned layout plans, the optimization of workstation layout is an important step. The question has arisen as to whether and how the known methods (e.g., CRAFT, CORELAP, MST, or Schmigalla triangles) will work well with 3D printing machines.

The concept of layout is very broad and there is no strict definition of it. This is presented and interpreted in various ways; one of the most popular definitions states [1] that "A facility layout is an arrangement of everything needed for the production of goods or delivery of services. A facility is an entity that facilitates the performance of any job." [2–5]. On the other hand, the factors that should be taken into account when designing the layout are clearly specified, ensuring adjustment to the requirements of the production system [6]:

- the type, variety, quality, and quantity of raw materials, work-in-process, and finished goods;
- the technological specification at each stage and the process of conversion as depicted in a flow process chart;

- the type of machinery and manufacturing aids necessary for the conversion process as stated above;
- the human factor of skill level at each stage, the number of persons required, the safety requirements and their ergonomics;
- the statutory and regulatory requirements as per provisions of the factories act must be considered for the plant layout;
- the line balancing and waiting factor decides the space requirement for the material storage;
- the material handling system is an integral part of the plant layout. It is one of the major cost areas. It is decided based on the extent of automation required and associated cost, speed, volume to be handled and type of material;
- the change factors in terms of future expansion, product modification, product range, flexibility and versatility of the organization, its products and technology; and
- service factors for machines and men have to be provided for. The space for machine maintenance, safety requirements for men and machines, provision of canteen, urinals, and toilets for workers, etc. is to be provided in the plant layout.

The variety of layout requirements, their large number and scale (from the location of the production hall on the industrial plot to the design of a single workstation), and the need to combine knowledge from various areas (from health and safety regulations, requirements regarding the order of arrangement of workstations, access between them, necessary connections, to their ergonomics) makes the implementation of a comprehensive layout project long and costly.

With the growing popularity of 3D printing techniques [7,8], layout designers will increasingly face the task of designing the layout of workstations taking into account the requirements for this type of device.

## 2. 3D Printing Methods—Information Essential for Designing Layout Plans

Additive manufacturing techniques offer great opportunities compared to traditional manufacturing methods such as, for example, foundry, plastic working, or plastic processing, allowing for the production of objects with complex geometries [9,10]. The model produced with the use of additive techniques was built by adding material layer by layer. The successively applied layers ultimately create the finished product. Various 3D printing methods can be used to apply the material. The following main types of additive technologies can be distinguished [11]:

- FDM (3D printing with the use of thermoplastics);
- SLA, DLP, MJP (using light-curing resins);
- SLS and MJF (3D printing with the use of powdered plastics);
- SLM, DMP, DMLS and EBM (3D printing with the use of powdered metals);
- CJP (3D printing with the use of gypsum powder); and
- LOM (3D printing with the use of foil or paper).

The most widespread and known techniques include SLA, SLS, and FDM. For further analysis, SLS and SLM devices were selected as basic devices for "factory building", based on the assumption that the offered products will use various materials (i.e., powdered plastics and metals).

### 2.1. Description of Selected Popular 3D Printing Methods

The SLA method (i.e., stereolithography) was the first of the developed rapid prototyping methods. The company 3D Systems patented this method in 1986 in the United States. In 1987, it began to manufacture machines using the technique of stereolithography [12]. The SLA method is based on layered polymerization of epoxy or acrylic resin with a laser beam. It is necessary to build structures that support the model, which take the form of thin rods and are usually removed mechanically [13]. The process begins by building a model in a 3D CAD system, which is then converted to the STL format. The prepared material in the form of a liquid resin is placed in the tub of the device. Before each layer

is hardened, the scraper smooths the sheets of liquid and removes air bubbles. The laser beam scans the areas that show the current cross-section of the created element, which causes polymerization. Then, the working plate is lowered by the thickness of the layer. The whole process is repeated until the final geometry of the manufactured item is obtained. The finished element must be cleaned of unbound resin. This is done by rinsing the product in isopropanol or acetone. Then, the supporting structures must be removed mechanically. The last step is supplementary UV irradiation so that the polymerization process is completed in the entire volume of the model [14]. The element prepared in this way can be subjected to additional finishing:

- grinding/smoothing, and
- varnishing.

Various types of resins are the materials used in the SLA method. Standard resin is mainly used to create concept models and prototypes [15,16]. ABS resin is used for elements that must exhibit high strength and significant elongation. The products that are to be elastically compliant are made of elastic resin. There are also resins such as high-temperature, foundry, or medical [17]. When designing a layout for rapid prototyping machines using the SLA method, it is necessary to take into account that apart from the printer itself, other devices are also used. The entire instrumentation also includes:

- UV irradiation chamber, and
- model rinsing bathtub.

The full name of the SLS method is selective laser sintering. This was developed and patented by the founders of one of the first 3D printing companies—Desk Top Manufacturing Corporation [18]. The SLS method is based on the layered solidification of materials that are used in the form of a powder. The individual layers are joined by a laser beam that affects the powder surface. The whole process begins with loading a 3D spatial CAD model, which is then converted by the software to the STL format, thanks to which the model is divided into individual layers, then control instructions for the machine are generated [19]. The prepared plastic powder is applied to the work area with a scraper or a roller. There, it is heated and then laser sintered. The areas that show the current cross-section of the model are scanned by the laser beam, which causes their fusion. Then, the working plate is lowered by the thickness of the layer and another dose of powder is applied. The whole process is repeated until the final geometry of the manufactured item is obtained [20]. The final product, which we receive after printing, should be cleaned of any residual powder. Additionally, depending on the requirements and needs, it can be subject to finishing [21]:

- smooth grinding (sandblasting),
- surface sealing, and
- varnishing.

The materials used in the SLS method are plastic powders. The most popular is the PA12 polyamide due to its high flexibility and high mechanical properties. Other materials used in this method include PA12 polyamide with the addition of glass balls, thanks to which the manufactured elements will show greater strength and stiffness to compression as well as slightly increased thermal resistance of the material and alumide (a mixture of polyamide powder and aluminum filings). The SLS method allows one to produce a finished product using one device, although other machines and devices are also necessary for the entire process, which should be taken into account when building the layout plan. Powder material must be properly prepared before it can be used in the process and its remains after printing can be reused after prior preparation. The elements of the instrumentation equipment include:

- control panel,
- control cabinet,

- cooler,
- unpacking and screening station,
- unpacking device, and
- cabin sandblaster.

Another increasingly popular technique of 3D printing is the SLM method—Selective Laser Melting. This technology is protected by several patents [22,23]. Using this method, elements from metal powders are produced. However, they exhibit lower strength and durability than parts made with traditional shaping techniques. The SLM method ensures repeatability of mechanical properties, which allows the use of parts produced in this way as elements of machines and devices. The SLM method consists in melting metal powder with a metal beam. The process begins with applying a layer of powder and then levelling it. Selected areas are melted by the laser beam. The next step in the process is to lower the working plate by the layer thickness and apply the powder layer again. The whole process is repeated until the finished model is obtained. The produced model should be cleaned of unmelted powder and most often subjected to finishing treatments, which include:

- CNC machining and milling, and
- varnishing.

The materials most often used in the SLM method include stainless steels, pure titanium, and its alloys as well as low-melting, zinc, copper, tool steel, silicon carbide, or aluminum oxide alloys. In machines used for SLM printing, it is necessary to use a protective gas in the working chamber and the type depends on the powder that will be used to print the element. The melting parameters of the powder must be selected in such a way that the overheating of the area that is irradiated with the laser is as little as possible, since too much heating could distort the resulting product [12]. When designing the layout plan, one has to take into account that in the SLM method, apart from the main machine that produces the finished model, other devices are also necessary and without them, the entire process could not take place. Such elements include, for example:

- control cabinet,
- cooler,
- unpacking and screening station,
- lift truck,
- cabin sandblaster, and
- unpacking device.

To test the methods of optimizing the placement of 3D printing devices, we planned to choose one of the several available methods. The selection was based on the popularity of the methods and their adequacy to the problem being solved.

*2.2. Characteristics of Methods for Optimizing the Arrangement of Workstations*

The classification of methods for optimizing the arrangement of workstations can be implemented, among others, according to the following criteria [24,25]:

- type of solution (exact and approximate);
- restriction of the choice of place (with or without restrictions);
- the way of presenting the shapes and dimensions of the stands (point methods—all devices have the same dimensions and modular methods—the size of square or triangular modules corresponds to the size of the devices);
- method of positioning workstations (stepwise and iterative);
- other (e.g., methods taking into account the outline of the shape of machines and devices; and genetic algorithms, expert systems, or solutions offered by producers of CAD programs).

When choosing a method to optimize the placement of stations for the planned research on the placement of 3D printing machines and devices, it was decided to take into account the most commonly used methods that simultaneously differed from each other in terms of the operation algorithm. The length of transport routes, the number of transport operations, transport costs, weight of transported materials, or the volume of transports can be used for optimization criteria. For the purposes of research, a total of five methods were selected for further analysis:

- CRAFT,
- Bloch-Schmigalla,
- ROC,
- MST, and
- CORELAP.

The CRAFT (computerized relative allocation of facilities technique) method was proposed in 1964 by E. Buff, G. Armour, and T. Vollmann. The CRAFT algorithm changes the positions of the stations in the initial layout to determine improved solutions based on the flow of materials; subsequent changes lead to the layout with the lowest total cost. CRAFT uses the material flow cost as a criterion to consider the best layout. The best design is the one that has a minimum total cost. CRAFT does not guarantee the least costly solution, as not all possible exchanges are considered. The quality of the final solution depends on the initial solution. Therefore, it is common practice to identify several different initial designs and try all the exchange combinations and then select the best solution generated [26–28].

In the CRAFT method, an existing layout or a completely new plan is considered as a block arrangement. The algorithm calculates the allocations of workstations and estimates the costs that will be incurred in the initial phase of the project. The impact on the cost measure is computed for two or more different position settings in the layout plan. The main goal of this method is to minimize the total cost of the TC function [16], which is defined by the formula:

$$TC = \sum_{i=1}^{n} \sum_{j=1}^{n} D_{ij} \cdot W_{ij} \cdot C_{ij}, \tag{1}$$

where $D_{ij}$ is the distance between station $i$ and station $j$; $W_{ij}$ is the volume of flow between station $i$ and station $j$; $C_{ij}$ is the cost of transport between station $i$ and station $j$; and $n$ is the number of stations.

The key steps of the algorithm are the calculation of the total cost of the TC function for the distance matrix describing the examined layout and the part flow matrix built on the basis of the sequence of technological operations and the demand for parts at different time periods.

The Bloch-Schmigalla method was initiated in the 1950s by W. Bloch. It uses a mesh of equilateral triangles in order to find the correct distribution of positions. The method was further developed and modified by H. Schmigalla [29]. The Schmigalla method of triangles begins by determining the order in which the workstations will be located and their distribution at individual nodes of a mesh composed of equilateral triangles. One should start with setting up a pair of stations with the highest flow intensity. It was assumed that the optimal system will be obtained when the W value of the objective function is the smallest possible. The function W is expressed as follows:

$$W = \sum_{i=1}^{n} \sum_{j=1}^{n} S_{ij} \cdot L_{ij} \rightarrow minimum, \tag{2}$$

where $S_{ij}$ is the amount of part flow between the stations $i$ and $j$; and $L_{ij}$ is the distance between the stations, which is always equal to the side length of the equilateral triangle of the mesh.

The next steps of the algorithm provide for the construction of a matrix of the sequence of technological operations to select later, from the matrix of connections, the objects connected with the highest flow intensity; from these objects, we start arranging the positions on the mesh of equilateral

triangles. A table of the intensity of connections between sites that have already been deployed and sites that have not yet been located helps in determining the location of subsequent objects.

The description of the ROC (rank order clustering) method was published in 1980 by J. King. ROC enables the arrangement of workstations by grouping them into manufacturing cells, taking into account families of parts classified according to technological similarity [30,31]. Its algorithm provides for the construction of a matrix of transport links and then, for each column of this matrix, its so-called binary weight is determined, according to the formula:

$$BW_j = 2^{m-j},\tag{3}$$

where $m$ is the number of machines and $j$ is the machine number. For the value of each row, the decimal binary equivalent is then calculated using the formula:

$$DE_i = \sum_{j=1}^{m} 2^{m-j} \cdot a_{ij},\tag{4}$$

where $a_{ij}$ is the relationship between element $i$ and position $j$, according to the matrix of transport connections. The effect of the ROC method is obtaining the arrangement of workstations with a job shop structure.

The modified spanning tree (MST) algorithm is based on the selection of adjacent pairs of workstations using a designated adjacency weight matrix. This method is similar to the spanning tree algorithm, which relies on a set of vertices and their connecting edges, where each edge with a separate weight connects two vertices. The vertices correspond to the workstations and the edges to the transport routes [32]. In the MST method, the adjacency weight matrix $f'_{ij}$ is built based on Equation (5) [1]:

$$f'_{ij} = \left(f_{ij}\right) \cdot \left(d_{ij} + 0.5 \cdot \left(l_i + l_j\right)\right),\tag{5}$$

where $f'_{ij}$ is the adjacency weight matrix; $f_{ij}$ is the flow matrix; $d_{ij}$ is the recommended distances between $i$ and $j$ machines; $l_i$ is the $i$ machine dimension; and $l_j$ is the $j$ machine dimension.

Equation (5) uses the product of the number of transport connections between workstations and the distance to be covered during the implementation of transport activities to determine the weight of the adjacency. It should be noted that the orientation of the machines is known in advance, or possibly presumed, which may be considered as a limitation of this method. As a consequence, it may also be necessary to prepare several variants of potential solutions that differ only in the orientation of machines and devices, and re-evaluate them by the MST algorithm.

From the adjacency weight matrix $f'_{ij}$, a pair of workstations connected by the highest weight are selected—they will be placed next to each other on the layout plan—a row and a column with a pair of devices already placed are plotted from the matrix. The next device is selected on the same principle, building a one-dimensional matrix of the order of arrangement of workstations.

The CORELAP (computerized relationship layout planning) method is one of the approximate methods. The use of this method is beneficial if there are various relationships between the objects being arranged that cannot be represented by one quantity [33].

The CORELAP method is based on determining the adjacency weight. The method consists of three stages: planning surfaces, their size and connections between them; calculating and designing CORELAP; and the final stage is drawing the layout plan. In order to define the relationship between the individual surfaces, each pair is assigned a symbol A, E, I, O, U, or X with a specific value, starting with "absolutely necessary" and ending with "undesirable". Relationships between positions recorded in the matrix are assessed by calculating the value of the proximity indicator (TCR). On this basis, the order of placing the positions is determined based on the six rules proposed in the CORELAP method. Depending on the adjacency method (fully contiguous, point contact, and non-contiguous,

for which the adhesion coefficient is 1, 0.5, and 0, respectively), subsequent stations are arranged. The method was first used as an algorithm for arranging rooms or departments in industrial plants.

The presented characteristics of the methods of optimizing the arrangement of workstations show that their application for 3D printing machines is virtually problem-free with one exception. A significant limitation is related to the need to take into account auxiliary devices, often not connected to the main 3D printing device by means of transport. In this case, most of the methods based on material flow analysis simply omit these devices, and this may result in the lack of space for their placement on the layout plan. The CORELAP method does not have this disadvantage.

### 2.3. Environmental Conditions for Machines and Devices for 3D Printing, Important for the Layout Plan

Machines and devices for 3D printing must be provided with appropriate environmental conditions, which should be taken into account when designing layout plans. In addition, access to various media is necessary: electricity, compressed air, cooling water, or protective gas. In addition to the main machines that print a given element, a number of devices are also used that support the entire process so additional space should be allocated for them. For research work, it was assumed that the 3D printing devices used in the newly built plant—along with appropriate auxiliary machines—will be the devices offered by EOS GmbH:

- FORMIGA P 110, using the SLS method [34], and
- EOS M 290, using the SLM method [35].

3D printing machines and devices have strictly defined environmental conditions during operation, and the requirements relate to the permissible temperature in the room and relative air humidity. Separate requirements apply to the storage environment for the powder used in the SLS method. The specification of the mains connection includes the values of voltage, its fluctuations and frequency, the necessary mains protection, and the type of connection. The specification of the compressed air connection includes its consumption, operating pressure, minimum, and maximum pressure, compressed air temperature, its quality (including water and oil content), and the type of connection. All these information and requirements must be considered by the layout plan designer [36]. The key information in developing layout plans, however, are the dimensions of the machines as well as their weight. The main dimensions of an exemplary SLS device are shown in Figure 1a,b.

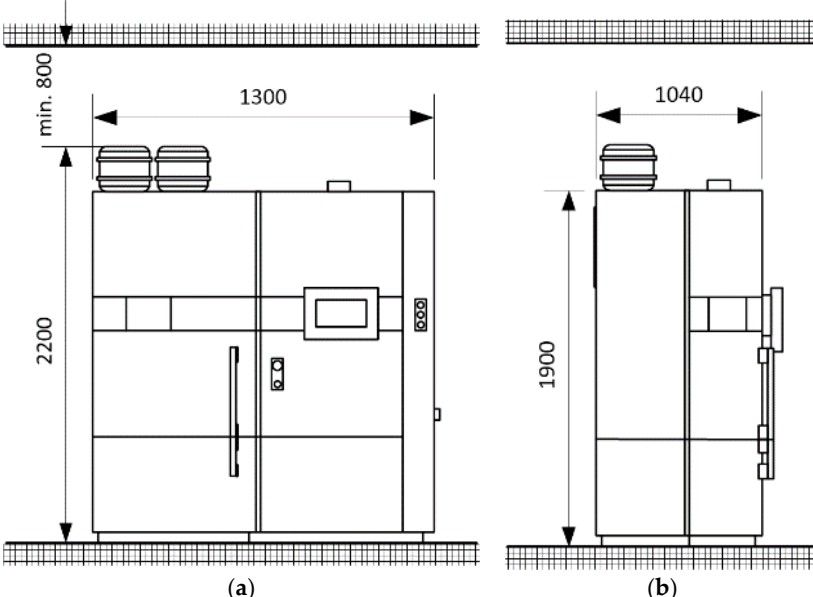

**Figure 1.** Main dimensions of an exemplary SLS device, dimensions in [mm]: (**a**) Front view of machine; (**b**) Side view of machine [34].

Manufacturers of 3D printing machines and devices also provide detailed information on the recommended distances from walls and other objects and the location of individual types of media, which is important for designers of layout plans (Figure 2).

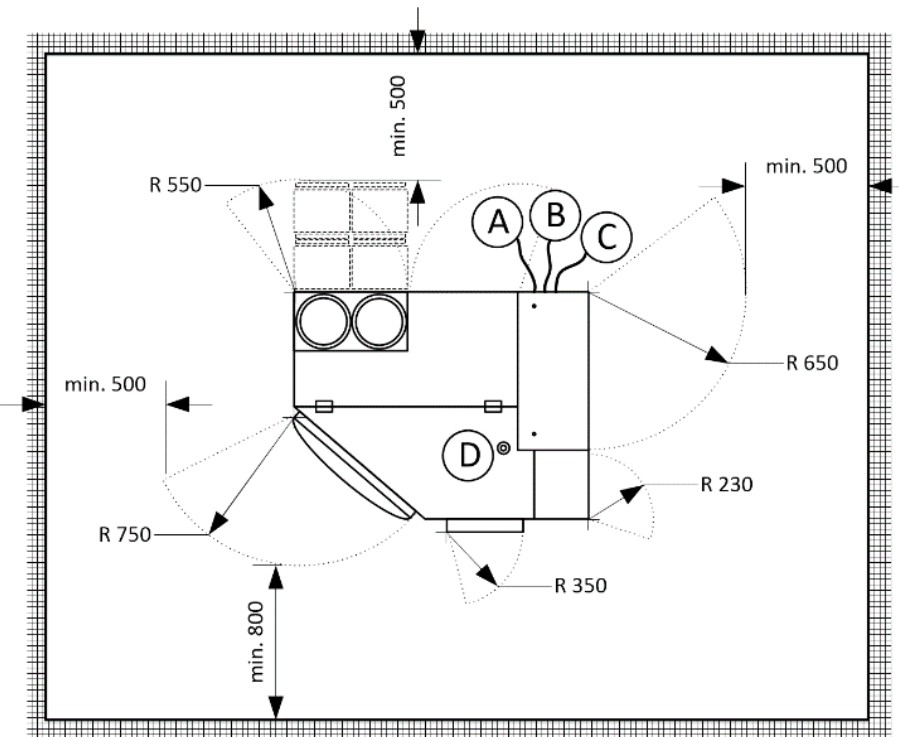

**Figure 2.** Requirements regarding space and location of connections for an exemplary SLS device [34].

3D printing devices that use a laser to sinter metal powder have specific requirements that must be met in order to ensure the correct operation of these machines. These requirements relate to the permissible temperature and humidity of the ambient air. During the SLM process, heat is released, which must be collected by a cooler operating in the system (e.g., air–water), hence, the requirements for the quality of the cooling water (e.g., pH value, chloride content, temperature, and minimum flow). An important role is also played by the shielding gas circuit filtering system (most often this gas is argon), which effectively collects the smelting contamination; a suitable place for it should be provided on the layout plan. During the operation of the SLM device, the noise emission at the level of approx. 70 dB (A) should be taken into account, which is also a factor that the designer of the layout plan cannot underestimate as the vicinity of a working 3D printing device can be troublesome in this respect. The power supply connection is particularly important for this type of 3D printing machine, due to the high energy demand of the laser sintering metal powder, the rated powers reach significant values so it is necessary to check these requirements with regard to the parameters of the power connection each time.

*2.4. Manufacturing Processes Implemented in SLS and SLM Technologies, Determining the Necessary Number of Machines*

For computational purposes, a hypothetical production plan was assumed; the production processes of two types of products will be carried out on 3D machines: product X manufactured in the SLS technique and product Y manufactured in the SLM technique. The production plan assumes the production of 20,000 pieces of product X and 23,000 pieces of product Y per year. Additional assumptions:

- number of products simultaneously printed on the working plate: Eight (for the SLS method) and 10 (for the SLM method);

- number of working days in a year: 250;
- number of production shifts: 2;
- the annual production plan for part X is 20,000 pcs, for part Y it is 22,800 pcs; and
- working day utilization factor (depends on the length of the work breaks, one break of 30 min is assumed): 0.9375.

To calculate the number of necessary devices in order to be able to implement the assumed production plan within the prescribed time, a formula was used that considered the ratio of the time necessary for technological operations to the available number of man-hours per year:

$$i_0 = \frac{t_{pz} + n \cdot t_j}{\psi_d \cdot D \cdot I \cdot 8} \tag{6}$$

where $t_{pz}$ is the unit time of a technological operation; $t_j$ is the preparation and completion time of a technological operation; $n$ is the number of manufactured items; $\psi_d$ is the working day utilization factor; $D$ is the number of working days per year; and $I$ is the number of shifts.

The form of the technological process for product X and Y is recorded in Tables 1 and 2, these tables present the designated number of machines according to Equation (6):

**Table 1.** Sequence of technological operations for product X with the calculated $i_0$ number of machines and devices necessary to implement the production plan.

| Operation No. | Device Name | Unit Time $t_j$ [min] | Preparation and Completion Time $t_{pz}$ [min] | Value $i_0$ | Number of Devices |
|---|---|---|---|---|---|
| 10 | Preparatory station | 10 | 20 | 0.889 | 1 |
| 20 | Mixing station | 15 | 20 | 1.333 | 2 |
| 30 | Sifter | 5 | 10 | 0.444 | 1 |
| 40 | SLS device | 320 | 30 | 3.556 | 4 |
| 50 | Cabin sandblaster | 5 | 10 | 0.444 | 1 |
| 60 | Grinder | 8 | 15 | 0.711 | 2 [1] |
| 70 | Varnishing station | 5 | 30 | 0.445 | 1 [1] |
| 80 | Packing | 7 | 10 | 0.622 | 2 [1] |

[1] Devices shared for the manufacturing process of products X and Y, $i_0$ values are then added up before rounding up.

**Table 2.** Sequence of technological operations for product Y along with the calculated $i_0$ number of machines and devices necessary to implement the production plan.

| Operation No. | Device Name | Unit Time $t_j$ [min] | Preparation and Completion Time $t_{pz}$ [min] | Value $i_0$ | Number of Devices |
|---|---|---|---|---|---|
| 10 | Feeding module | 15 | 10 | 1.520 | 2 |
| 20 | Filling module | 20 | 20 | 2.027 | 3 |
| 30 | SLM device | 430 | 35 | 4.375 | 5 |
| 40 | Microsand blaster | 10 | 10 | 1.013 | 2 |
| 50 | Grinder | 8 | 15 | 0.811 | 2 [1] |
| 60 | Varnishing station | 5 | 30 | 0.507 | 1 [1] |
| 70 | Packing | 7 | 10 | 0.709 | 2 [1] |

[1] Devices shared for the manufacturing process of products X and Y, $i_0$ values are then added up before rounding up.

The calculated number of 3D printing machines and devices, along with their dimensions, will be the key information at the next stage of developing layout plans.

## 3. Results—The Effects of the Arrangement Optimization of 3D Printing Stations

To optimize the arrangement of workstations, it was decided to use the MST method, based on the adjacency weight matrix, built on the basis of material flows. With this algorithm of procedure, auxiliary stations, characteristic for 3D printing machines and devices, are not taken into account. Thanks to this, it will be possible to collect information on whether this limitation is a significant disadvantage of this type of optimization method in the case of 3D printing devices.

### 3.1. Results of Optimization of the Arrangement of Workstations

For the purposes of the calculations, the MST method assumed that the size of the transport batch would correspond to the number of parts on the platforms of SLS and SLM devices. Machines will be located in a way providing, apart from the minimum recommended distances between them, access to the transport road for each of them. It was assumed that a hand truck would be used to transport a batch of elements. It will also be necessary to provide additional space for buffers (intermediate storage areas). The minimum recommended distances depend on the requirements of the manufacturers or on the size of the machines, the type of adjacency (side of the device, rear of the device, side of the machine where the operator works in relation to the sides of another machine, wall or transport road, etc.), and the dimensions of the machines.

Table 3 contains the information matrix of the part flow for the assumed technological process of products X and Y. The sequence of technological operations to be implemented is recorded in this matrix. It also includes the calculated number of transport activities, based on the number of manufactured products and the number of pieces in the transport batch.

The next step in the MST method is the development of the flow matrix (Table 4), containing information on the number of transport activities connecting workstations into subsequent pairs. Certainly, these calculations are based on the part flow information matrix. The numbering of workstations from Table 3 was also retained in the following tables (Tables 4–8).

Table 5 contains the selected distances between the stations connected by transport activities, depending on the method of the assumed adjacency (in this case "side to side") or the requirements taken from the technical and commissioning documentation provided by the manufacturer of the device.

The selected distances between the devices in Table 5 also depended on the dimensions of the machines (Table 6), where the larger the dimensions of the machines, the greater the recommended distances between them.

The values in the adjacency weight matrix $f'_{ij}$, presented in Table 7, were calculated on the basis of the previously presented Equation (5) from Section 2.2.

The result of the MST method is the sequence of workstations, read from the adjacency weight matrix $f'_{ij}$, written in the characteristic form of a single-line matrix. For the analyzed case, the results are presented in Table 8.

The pair of Output Warehouse (14) and Packing (13) stations showed the greatest weight of adjacency, therefore the pair from which the reading of the sequence of the arrangement of workstations began. The Output Warehouse (14) was the last station in the chain, therefore subsequent stations were selected from the adjacency weight matrix preceding the Packing (13) station, guided by the calculated weight for transport activities.

**Table 3.** Part flow information matrix.

| | Machine | | | | | | | | | | | | | | | Part Demand | Batch Size | Number of Batches |
|---|---|---|---|---|---|---|---|---|---|---|---|---|---|---|---|---|---|---|
| | Input Warehouse | Preparatory Station | Mixing Station | Sifter | SLS Device | Cabin Sandblaster | Feeding Module | Filling Module | SLM Device | Microsand blaster | Grinder | Varnishing Station | Packing | Output Warehouse | | | | |
| | 1 | 2 | 3 | 4 | 5 | 6 | 7 | 8 | 9 | 10 | 11 | 12 | 13 | 14 | | | |
| Part X | 1 | 2 | 3 | 4 | 5 | 6 | | | | | 7 | 8 | 9 | 10 | 20,000 | 5 | 4000 |
| Part Y | 1 | | | | | | 2 | 3 | 4 | 5 | 6 | 7 | 8 | 9 | 22,800 | 6 | 3800 |

**Table 4.** Flow matrix $f_{ij}$.

| | | Machine | | | | | | | | | | | | |
|---|---|---|---|---|---|---|---|---|---|---|---|---|---|---|
| Machine | | 1 | 2 | 3 | 4 | 5 | 6 | 7 | 8 | 9 | 10 | 11 | 12 | 13 | 14 |
| | 1 | | 4000 | | | | | 3800 | | | | | | | |
| | 2 | | | 4000 | | | | | | | | | | | |
| | 3 | | | | 4000 | | | | | | | | | | |
| | 4 | | | | | 4000 | | | | | | | | | |
| | 5 | | | | | | 4000 | | | | | | | | |
| | 6 | | | | | | | | | | | 4000 | | | |
| | 7 | | | | | | | | 3800 | | | | | | |
| | 8 | | | | | | | | | 3800 | | | | | |
| | 9 | | | | | | | | | | 3000 | | | | |
| | 10 | | | | | | | | | | | 3800 | | | |
| | 11 | | | | | | | | | | | | 7800 | | |
| | 12 | | | | | | | | | | | | | 7800 | |
| | 13 | | | | | | | | | | | | | | 7800 |
| | 14 | | | | | | | | | | | | | | |

**Table 5.** Clearance matrix $d_{ij}$ for the analyzed case.

| | | Machine | | | | | | | | | | | | |
|---|---|---|---|---|---|---|---|---|---|---|---|---|---|---|---|
| Machine | | 1 | 2 | 3 | 4 | 5 | 6 | 7 | 8 | 9 | 10 | 11 | 12 | 13 | 14 |
| | 1 | | 0.90 | | | | | 0.90 | | | | | | | |
| | 2 | | | 0.40 | | | | | | | | | | | |
| | 3 | | | | 0.40 | | | | | | | | | | |
| | 4 | | | | | 0.50 | | | | | | | | | |
| | 5 | | | | | | 0.50 | | | | | | | | |
| | 6 | | | | | | | | | | | 0.50 | | | |
| | 7 | | | | | | | | 0.40 | | | | | | |
| | 8 | | | | | | | | | 0.50 | | | | | |
| | 9 | | | | | | | | | | 0.50 | | | | |
| | 10 | | | | | | | | | | | 0.50 | | | |
| | 11 | | | | | | | | | | | | 0.40 | | |
| | 12 | | | | | | | | | | | | | 0.40 | |
| | 13 | | | | | | | | | | | | | | 0.90 [1] |
| | 14 | | | | | | | | | | | | | | |

[1] All selected distances between devices are expressed in meters.

**Table 6.** Machine lengths $l$ for the analyzed case.

| Machine Lengths $l$ [m] | | | | | | | | | | | | | |
|---|---|---|---|---|---|---|---|---|---|---|---|---|---|
| 1 | 2 | 3 | 4 | 5 | 6 | 7 | 8 | 9 | 10 | 11 | 12 | 13 | 14 |
| 5.00 | 1.13 | 0.68 | 0.65 | 1.75 | 1.26 | 0.80 | 0.80 | 1.57 | 0.75 | 1.55 | 1.40 | 1.30 | 5.00 |

**Table 7.** Adjacency weight matrix f′$_{ij}$.

| | Machine | | | | | | | | | | | | | |
|---|---|---|---|---|---|---|---|---|---|---|---|---|---|---|
| | 1 | 2 | 3 | 4 | 5 | 6 | 7 | 8 | 9 | 10 | 11 | 12 | 13 | 14 |
| 1 | | 15,850 | | | | | 14,440 | | | | | | | |
| 2 | | | 5210 | | | | | | | | | | | |
| 3 | | | | 4260 | | | | | | | | | | |
| 4 | | | | | 6800 | | | | | | | | | |
| 5 | | | | | | 8020 | | | | | | | | |
| 6 | | | | | | | | | | | 7,620 | | | |
| 7 | | | | | | | | 4560 | | | | | | |
| 8 | | | | | | | | | 6403 | | | | | |
| 9 | | | | | | | | | | 6308 | | | | |
| 10 | | | | | | | | | | | 6270 | | | |
| 11 | | | | | | | | | | | | 14,625 | | |
| 12 | | | | | | | | | | | | | 13,650 | |
| 13 | | | | | | | | | | | | | | 31,590 |
| 14 | | | | | | | | | | | | | | |

**Table 8.** Designated sequence of workstations for the analyzed case.

| Computational Sequence of Workstations | | | | | | | | | | | | | |
|---|---|---|---|---|---|---|---|---|---|---|---|---|---|
| 10 | 9 | 8 | 7 | 1 | 2 | 3 | 4 | 5 | 6 | 11 | 12 | 13 | 14 |

### 3.2. Developed Layout Plan

When developing the layout plan, the information collected thus far regarding workstations, the required and recommended distances between them, their number and dimensions, and the location determined according to the MST method were used. In addition, it was assumed that the shape of the plot on which the production hall is located enables the delivery of the necessary raw materials and other consumables on one side of the production hall and the receipt of finished products on the other. With this location of the Input and Output Warehouses, it was easier to ensure non-intersecting material flows. Access to the transport road was ensured via inter-operational buffers for batch-based transport operations. The developed layout plan, taking into account the above requirements, is shown in Figure 3.

A certain difficulty in the design process was the transfer of the designated layout of workstations from the MST method to a specific layout plan, due to the need to consider technological limitations related to:

- taking into account the required/recommended minimum distances between devices, not only "side to side", but also clearances (e.g., from the transport road or walls);
- the need to adjust to the shape and dimensions of the production hall;
- limited availability of the required connections for machines and devices in the production hall; and
- the need to minimize routes for operators between devices and buffers, in particular in cases where there are several identical devices.

The social part for employees was not included in the designed production hall in order to limit the size and detail of the drawing and thus increase its readability. For the same reason, the dimensions of machines and devices or the assumed distances between them are not shown.

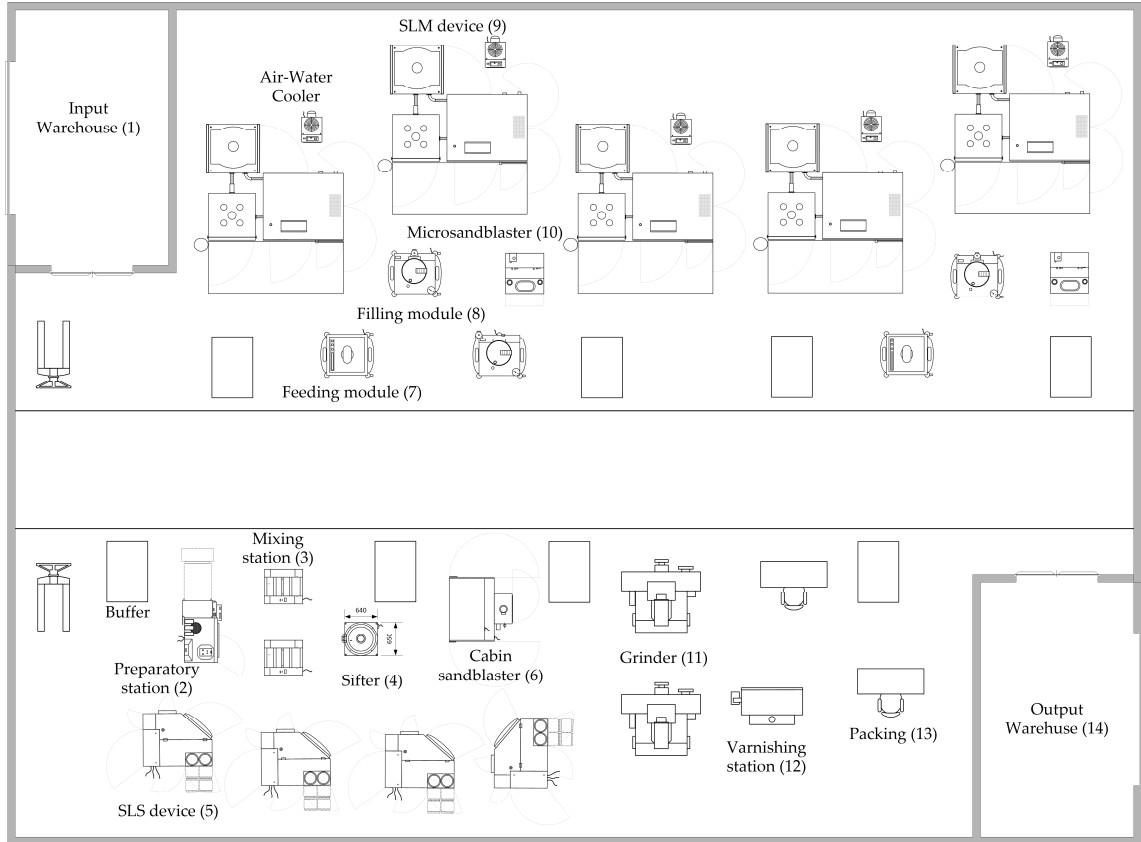

**Figure 3.** Sample layout plan design for 3D printing machines and devices.

## 4. Discussion—Guidelines for the Placement of 3D Printing Devices

In general, mathematical methods of optimizing the arrangement of workstations work well when using 3D printing machines; special attention should be paid to the following aspects when designing layout plans:

- there is a problem with the arrangement of auxiliary devices for 3D printing machines since some of them are not covered by materials or product transport, and therefore they are not included in the methods of optimizing the arrangement based on the number of transport activities. Certainly, they should be placed on the layout plan, which often distorts the optimal—under given conditions—solution. Such devices include control cabinets, coolers of various types or filtration systems;

- the calculated load of 3D printing machines shows that a significant part of the auxiliary devices is used to a small extent, the solution is to share these devices, which complicates the task of the layout plan designer;

- special attention should be paid to the connections as they differ depending on the 3D printing method;

- intermediate storage areas are most often not included in mathematical methods of arranging workstations, but they are always placed as close to machines and devices as possible in order to shorten the distance covered by operators as much as possible; and

- recommended distances between machines, available in the literature on the subject are usually much smaller than the requirements of manufacturers of specific 3D printing devices and the limitations resulting from the need to maintain access to the service doors.

A significant difficulty is also the process of transforming the solution from a mathematical method of optimizing the arrangement of workstations to a specific layout since the limited space, dimensions

of workstations, and the required distance to the transport route and the need to connect utilities must be taken into account. This design stage is only partially supported by CAD software, which provides ready-made machines and devices as parameterized objects, so it relies on the knowledge and experience of the layout designer.

**Author Contributions:** Conceptualization, A.K. and R.W.; Methodology, A.K.; Validation, R.W.; Formal analysis, A.K.; Resources, A.K. and R.W.; Data curation, A.K.; Writing—original draft preparation, A.K. and R.W.; Writing—review and editing, A.K.; Visualization, A.K. and R.W.; Funding acquisition, R.W. All authors have read and agreed to the published version of the manuscript.

**Funding:** This research received no external funding.

**Conflicts of Interest:** The authors declare no conflict of interest.

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
