# Peer review of "Layout Guidelines for 3D Printing Devices"

_applsci, doi:10.3390/app10186333_

Round 1

Reviewer 1 Report

Dear authors,

The manuscript "Layout guidelines for 3D printing devices" presents a mathematical method to optimize the layout of workstations featuring machines and devices with additive technology. I consider that this work is innovative and presents a very interesting approach that will cooperate for the implantation of these technologies in the industrial sector. My recommendation is accept after minor revision. I send you some suggestions or questions:

Introduction => explain better the advantage of this work, what novelty it can bring? 

Table 3 => it is not clear, please, explain better!

Author contributions => Please, verify! 

Some sentences show problems with formatting. Please, make the corrections. 

Reviewer 2 Report

The concept presented in the article of considering aspects related to the principles of system design for additive technology devices is undoubtedly very current. It is assumed that relatively new in terms of concept production systems that take into account mathematical methods of optimization will be implemented for 3D printing machines and equipment. The attempt to apply the selected mathematical method to optimize the layout of workstations using machines and to develop an exemplary layout of the workstation presented in the article provides the basis for establishing requirements and guidelines for the necessary space, installation and connections.

A synthetic review and comparison of the most known 3D printing methods was made, indicating the possibilities of developing a production system, and the characteristics of the methods of optimizing the layout of workstations was made. Although the research part is not very extensive, it presents potential directions of development in the area of optimization of the distribution of technological infrastructure.

Reviewer 3 Report

The introduction section can be improved with the state-of-the-art.

-The main question addressed by the research is dedicated to the 3d printing methodology concentrated on the development of the layout, which is the basis for the 3d printing process. -The paper is relevant and interesting because of the rapidly evolving usage of 3d printing in the critical areas - aerospace, military, healthcare, household. - The topic is original. However, the introduction section should be improved in order to provide more deep analysis of the related works. - The introduction section should be improved in order to provide more deep analysis of the related works.  - The paper is well structured and well written with a good level of English language.
-
The text is easy to read with the upper-intermediate level of English knowledge. - The conclusions section represents a summary of the experimental findings presented in the paper.    
